# Determinants of the Limpopo Province of South Africa’s Response to COVID-19: A Mixed Methods Protocol to Analyze and Share Lessons Learned

**DOI:** 10.3390/healthcare10050926

**Published:** 2022-05-18

**Authors:** Sogo F. Matlala, Takalani G. Tshitangano, Musa E. Setati, Naledzani J. Ramalivhana, Peter M. Mphekgwana

**Affiliations:** 1Department of Public Health, University of Limpopo, Sovenga, Polokwane 0727, South Africa; 2Department of Public Health, University of Venda, Thohoyandou 0950, South Africa; takalani.tshitangano@univen.ac.za; 3Department of Health, University of Limpopo, Polokwane 0700, South Africa; musasono@yahoo.com; 4Department of Public Health Medicine, University of Limpopo, Sovenga, Polokwane 0727, South Africa; 5Department of Pharmacy, University of Limpopo, Sovenga, Polokwane 0727, South Africa; drramalivhana.c99986@gmail.com; 6Research Administration and Development, University of Limpopo, Sovenga, Polokwane 0727, South Africa; peter.mphekgwana@ul.ac.za

**Keywords:** secondary data, COVID-19, lessons learned

## Abstract

COVID-19 is a priority health research agenda item in South Africa. The World Health Organization declared the COVID-19 outbreak as a Public Health Emergency of International Concern, which requires all countries to respond and share data with others. Responses included the implementation of measures to reduce the spread of COVID-19 and protect health systems from being overwhelmed by seriously ill patients. Each country was mandated to assess its own risk and rapidly implement the necessary measures to reduce COVID-19 transmission and minimize its impact. Countries were further encouraged to share their experiences of responding to the COVID-19 pandemic. Media reports in South Africa suggest that the Limpopo Department of Health implemented a successful COVID-19 response. This study aims to analyze, document and publish those successes to make them accessible to other researchers and public health practitioners. The study will also allow for the participation of public health students to meet the requirements of their postgraduate degrees. This convergent parallel mixed method study will collect secondary data of responses to COVID-19 by the Limpopo Department of Health from the records the department keeps. Quantitative and qualitative data detailing activities and statistics describing facilitators and barriers to implementation of COVID-19 response from March 2020 will be extracted from records.

## 1. Introduction

Coronavirus disease (COVID-19) is a disease caused by a virus called severe acute respiratory syndrome coronavirus 2 (SARS-CoV-2) [1]. Although it is related to severe acute respiratory syndrome (SARS) and the Middle East respiratory syndrome (MERS) which all results from infection with a coronavirus, COVID-19 is different regarding community spread and severity [2]. Furthermore, the World Health Organization (WHO) declared the COVID-19 outbreak as a Public Health Emergency of International Concern (PHEIC) which requires all countries to prepare for containment, prevention of spread and sharing of data with other countries [3]. WHO urged countries to implement measures to slow the spread of COVID-19 and protect their health systems from being overwhelmed by seriously ill patients. Countries were further encouraged to share their experiences of responding to COVID-19, since there is no one-size-fits-all approach. As such, each country was mandated to assess its own risk and rapidly implement the necessary measures to reduce COVID-19 transmission and minimize its impact.

Media reports in South Africa suggest that Limpopo Department of Health (LDoH) is implementing a successful COVID-19 response [4,5,6,7,8]. It is important for these implementation successes to be researched and documented in forms that are accessible to researchers, policy makers, health systems managers and other stakeholders to facilitate their replication and sharing of lessons. Sharing these successes (and possibly some failures) with the research community makes it possible for them to be replicated for the benefit of other provinces in South Africa as well as other countries in the world. Therefore, this study aims to analyze and report about the response of LDoH to COVID-19. LDoH collects data and records activities as it responds to the COVID-19 pandemic. However, these successes (and possibly some failures) are not accessible to the research community. This study will thus analyze and document those successes in a way that they are accessible to the research community.

### 1.1. Definition of Concepts

#### 1.1.1. COVID-19 Response

WHO, in its 2008 publication titled Communicable disease alert and response for mass gatherings, defines a response as “actions taken before, during and immediately after the occurrence of a disaster, to ensure that the effects of that disaster are minimized and people are given immediate relief and support” [9] (p. 11). Applying this definition to COVID-19, response refers to activities taken before and during the outbreak of this current pandemic to ensure that its effects are minimized and people are given immediate relief and support. In this study, COVID-19 response refers to actions taken by LDoH to minimize COVID-19’s effects and provide immediate relief and support to communities.

#### 1.1.2. Determinants of Healthcare Practice

The determinants of healthcare practice refer to things that may support or prevent the implementation of healthcare interventions [10]. In this study, the determinants of healthcare practice refer to facilitators and barriers to the implementation of COVID-19 response activities by LDoH.

#### 1.1.3. Lessons Learned

Lessons learned refers to recorded details of activities and statistics describing both the positive and negative process of implementing interventions [11]. In this study, lessons learned refers to a record of activities and statistics describing facilitators and barriers to implementation of COVID-19 response activities by LDoH.

#### 1.1.4. Preparedness

Preparedness refers to a continuous cycle of planning, organizing, training, equipping, exercising, evaluating and taking corrective action during a disaster [12]. In this study, preparedness refers to all LDoH’s activities that were planned and implemented during the South African COVID-19 lockdown.

#### 1.1.5. Readiness

Readiness refers to the action and effect of being prepared to respond to an emergency [13]. In this study, readiness refers to all LDoH’s information contained in reports and means of verification provided as evidence of readiness.

### 1.2. COVID-19 Preparedness and Response

COVID-19 preparedness and response indicate important roles that all sectors of society including individuals, families and communities must play in alleviating the effects of this pandemic [14]. The development of the ability to respond, planning, coordination and communication activities are cross-cutting issues which require integrated actions by all sectors. The national government should lead the overall coordination and communication efforts and channel most activities through the health sector. Provincial departments should also plan and coordinate activities at a provincial level as well as monitor and evaluate the impact of those interventions. The health sector has a primary responsibility to raise awareness of the risks and potential health consequences of the COVID-19 pandemic. Furthermore, the health sector (both public and private healthcare services) should show leadership and advocacy by guiding the response efforts of other sectors in the country. WHO urges countries to take the necessary actions to further slow the spread of COVID-19 and protect their health systems from becoming overwhelmed by seriously ill patients who may need admission into an intensive care unit (ICU).

WHO has issued the COVID-19 Strategic Preparedness and Response Plan (SPRP) for 2021 to guide coordinated actions that must be taken at national, regional, and global levels to overcome the ongoing challenges brought by COVID-19 [14]. The SPRP has the following aims:To slow and stop transmission, prevent outbreaks and delay spread;To provide optimized care for all patients, especially those who are seriously ill;To minimize the impact of COVID-19 on the health systems, social services and economic activities.

According to the WHO, important lessons have been learned so far but there is still much to learn about COVID-19 and its impact on the society [2]. It is therefore important for researchers to continuously generate and share knowledge as preparedness, readiness and response actions are to be driven by rapidly accumulating scientific and public health knowledge. Although countries are urged to share experiences, each countries should know that there is no single response to COVID-19 that can be applied at all countries without modification. Each country should assess its own risk and quickly implement appropriate measures to reduce COVID-19 transmission and minimize its impact on the society. As such, South Africa has adopted a staged response to COVID-19 which is made up of the following eight stages:Stage 1: Preparation;Stage 2: Primary prevention;Stage 3: Lockdown;Stage 4: Surveillance and active case-finding;Stage 5: Hotspots;Stage 6: Medical Care (for the peak);Stage 7: Bereavement and the aftermath;Stage 8: Ongoing vigilance.

South Africa has nine provinces with each province having its own Legislature, Premier and Executive Council, and therefore, each has the capacity to formulate its own relevant response to COVID-19 which has to be in line with that of the country as a whole. There is a National COVID-19 Command and Control Council (NCCC) at the national government level to coordinate responses to the COVID-19 pandemic and each province has a Provincial COVID-19 Command and Council (PCCC) which has to approve all COVID-19 health activities at provincial level [15,16]. As Limpopo Province is one of the nine provinces of South Africa, this study aims to explore how the province is responding to COVID-19 guided by the staged response adopted by the country.

## 2. Materials and Methods

### 2.1. Aim and Objectives

This protocol is for a study that aims to describe LDoH’s response to COVID-19 and share lessons learned. The objectives are:To determine health facility preparedness to respond to COVID-19 in Limpopo Province;To determine COVID-19 in-patient case fatality rate in Limpopo Province;To determine factors contributing to COVID-19 in-patient case fatality rate in Limpopo Province;To explore strategies to reduce COVID-19 case fatality rate in Limpopo Province;To explore the impact of COVID-19 strategic interventions on the general health sector and the workforce;To explore challenges faced by the LDoH regarding COVID-19 response.

### 2.2. Research Question

How did LDoH prepare and respond to COVID-19 and what lessons can be learned?

### 2.3. Research Methodology

The study will use secondary analysis which is a research method that involves analysis of secondary data. Secondary data is existing data that was originally collected by other people, researchers or organizations for their own primary purposes [17,18]. In some cases, researchers do not use all of the data they collected, and the unused data becomes available as secondary data for other researchers to use in answering their own research questions. Availability of secondary data is helpful to researchers who are constrained for time and other resources. The use of secondary data expedites research as it eliminates costs and other lengthy processes of collecting primary data. MacInnes [19] indicates that the use of secondary data in research avoids unnecessary duplication of research effort as the collection of primary data requires great effort. In research on COVID-19, information is constantly changing [20], hence using secondary data allows research to be completed and findings to be produced quicker. This facilitates the cumulative growth of knowledge to inform policy and practice.

Parisot et al. [21] indicate that critics regard secondary data as data that was generated without the current researchers’ interventions as it was collected without reference to the current research question. It is thus important for researchers to follow a rigorous process when using secondary data. Such a rigorous process includes the development of a research question as well as identification and evaluation of relevant data. Johnston [17] suggests that secondary data should be evaluated by following a process which checks the original purpose of collecting primary data, the integrity of the data collection process, the type of data collected and the period during which data was collected in order to determine its suitability to the current research project.

Obtaining secondary data for health research requires formal approvals from gatekeepers to access documents which are kept under strict data protection regulations to protect confidential information and thus respect the privacy of people and institutions. Secondary data can be analyzed quantitatively and qualitatively, making them suitable for a mixed method research project [18].

### 2.4. Research Design

The study will use a mixed method approach to collect secondary data in response to COVID-19 from LDoH. A mixed method involves the collection of both quantitative and qualitative data. Secondary data can be analyzed either quantitatively or qualitatively, as such this study will use a convergent parallel mixed method design [22]. In a convergent parallel design, the quantitative and the qualitative components are conducted concurrently and both components are given equal attention. Data for each component is collected and analyzed independently. Results are then interpreted together to arrive at a conclusion. The second and third objectives will adopt a quantitative method, whereas the other four objectives will be studied using a qualitative method.

### 2.5. Population and Sampling

The population of this study will be all the LDoH COVID-19 preparedness and response reports from March 2020 that bears information to assist the researcher to achieve the objectives of this study. The study will include all LDoH’s COVID-19 preparedness and response reports from March 2020.

### 2.6. Data Collection and Analysis

Quantitative and qualitative data will be extracted from records of activities and statistics describing facilitators and barriers to implementation of COVID-19 response by LDoH. An appropriate data extraction tool will be designed to collect quantitative and qualitative data to meet specific objectives of the study. The tool will have two sections, with the first section collecting quantitative data on health facilities preparedness and case fatality rates, while the second section will collect qualitative data on strategies to reduce the case fatality rate, the impact of strategic interventions, as well as challenges experienced by LDoH. To collect qualitative data for the second section, the following questions will be separately asked:What are the strategies used by LDoH to reduce case fatality rate in health facilities and households of the province?What is the impact of COVID-19 strategic interventions on the health sector and the workforce?What are the challenges faced by the LDoH regarding COVID-19 response?

Descriptive and inferential analyses will be used to determine case fatality rate and its determinants while qualitative content analysis and thematic analysis will be used to explore strategies to reduce case fatality as well as challenges faced by LDoH when responding to COVID-19 [23,24].

### 2.7. Validity, Reliability and Trustworthiness

Validity will be ensured by conducting a literature review on the assessment of facilitators and barriers to the management of COVID-19 to inform the data extraction tool. Furthermore, validity will be ensured by presenting the data extraction tool to experts in the Departments of Public Health at the University of Venda (UNIVEN) and University of Limpopo (UL) as well as Department of Public Health Medicine at UL where the researchers are based. Reliability will be established by accurate and careful phrasing of items in the data extraction tool to avoid ambiguity. Additionally, a pilot study will also be performed to pre-test the tool as De Vos et al. [25] suggest.

Trustworthiness of the qualitative data will be based on the model of Lincoln and Guba criteria, namely credibility, transferability, dependability and confirmability [26]. Credibility will be achieved through triangulation by using qualitative and quantitative approaches. In addition, this protocol has been evaluated by a research ethics committee who gave their opinions regarding the ability of the research methodology to yield credible results (and they have issued a clearance certificate). Transferability will be ensured by giving detailed descriptions of data collection and analysis processes to allow judgments about transferability to be made by readers. To ensure dependability, an in-depth description of the data collection, findings, interpretations and recommendations will be achieved in all publications emanating from this protocol in order to confirm that the findings are supported by data and there is internal coherence. In addition, the researchers will analyze data independently, then discuss their findings and agree on the final themes to ensure confirmability.

### 2.8. Ethical Clearance and Permission to Conduct the Study

Ethical clearance was requested from Turfloop Research Ethics Committee (TREC) at UL and was granted (TREC/293/2021: IR). Gatekeeper permission to conduct the study was requested from Limpopo Province Department of Health. This was achieved after obtaining ethical clearance from TREC. The request for gatekeeper permission is achieved by uploading the necessary documents which includes the research proposal, ethical clearance certificate and a covering letter into the National Health Research Committee (NHRC) portal. Gatekeeper permission was granted (Ref: LP_2021-11-017). Further gatekeeper permission will be sought should doctoral students as well as Master of Public Health (MPH) and Master in Medicine (Mmed in Public Health Medicine) students from either UL or UNIVEN wish to make use of the available COVID-19 data to fulfil requirements for their degrees.

### 2.9. Confidentiality, Anonymity and Privacy

Confidentiality, anonymity and privacy of individual facilities and health workers will be protected. Care will be taken to remove any information that can lead to identification of individual patients and health workers during the publication of the findings. Data obtained will only be used for research purposes as outlined in this protocol and will not be shared with unauthorized people. Maintaining confidentiality, anonymity and privacy will be possible as all researchers have both ethical and professional integrity.

### 2.10. Informed Consent and Protection from Harm or Risk

This study does not require informed consent from participants as it will use secondary data. As indicated above, permission to access and use data has been obtained from LDoH. The study will respect the privacy and confidentiality of patients, communities, institutions and health workers whose data will be used as explained above. As secondary data will be used, there is no risk of exposure to COVID-19.

## 3. Results

Ethical clearance (TREC/293/2021: IR) and gatekeeper permission (Ref: LP_2021-11-017) have since been granted; data analysis has also started. As the research project intents to include interested public health students from UL and UNIVEN, recruitment will start through faculty of health sciences at both universities. Results will be shared with LDoH policymakers, other key stakeholders and health care facilities staff.

## 4. Discussion

COVID-19 is a priority health research item for the National Department of Health as well as LDoH. Results and lessons learned will be shared through conference presentations and publications in peer-reviewed and accredited conference proceedings, journals and book chapters. Sharing of findings may help other provinces nationally and internationally to adopt and improve on the successes and also shortcomings of LDoH. MPH students from UNIVEN and UL as well as those studying Master in Medicine (Mmed in Public Health Medicine) may access data and be supervised by the researchers attached to the two universities. Although the researchers will guard against breaches of anonymity, this cannot be totally guaranteed as someone with sufficient knowledge of the context might be able to know the identity of either participants or communities involved.

## Data Availability

Not applicable.

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
