# Peer review of "Determinants of the Limpopo Province of South Africa’s Response to COVID-19: A Mixed Methods Protocol to Analyze and Share Lessons Learned"

_healthcare, 2022, doi:10.3390/healthcare10050926_

Round 1

Reviewer 1 Report

The paper proposal for  research study protocol, concerning the determinants of Limpopo Province SA for its response to theCovid-19 pandemic is of high interest. 

There is some need for epidemiology to reach more research protocols. I would encourage the authors to go forward.

 The research protocol itself is well written, relatively and easy to understand.

There are just some minor aspects I would suggest to improve the paper.

First, the abstract is not that clear. It does not reflect the structure and the arguments developed in the paper. It does not reflect either its richness (but here there is a trade off not to augment the abstract much). 

The use of the expression “mixed methods” protocol is not clear (use in the title) in the abstract you mention mixed methods study. For me, and according to the literature, mixed-method is about combining and integrating in a research project quantitative with qualitative data and methods.

In the paper you mentioned the failures - but not in the abstract or in the first paragraph of the introduction.

In section 1.1.3 Lessons learned, I would ask if a more detailed presentation or discussion of facilitators and barriers could improve the clarity and information given in this protocol. 

There is an important point here - and elsewhere in the paper, for example in the discussion, or the research design - that I find important. It is about the context of the research and protocol. Not only the overall context of the Limpopo province, but the various data gathered in relation to the underlying population.  

I would like to know if you consider a detailed or at least systematic on some counts of the way the information was gathered. (see also how the information selected the patients for example in relation to the whole population of the population in the same geographical or social conditions, etc.).

Did you consider alternative measures (see page 4, 2.1 and 2.3

I would suggest to consider in 2.10 and in the discussion (maybe) the risks related to breach of privacy or anonymity. For example, some studies might guarantee the anonymity of individuals but not necessarily a community, if someone is sufficiently knowledgeable of the context of the gathering of the data. 

Is there any information on the protocol with a proper URL or portal that centralizes its use and its updated information?

Reviewer 2 Report

Dear editor

I understood that the journal publishes clinical trial protocols. In view of this, I believe that the material submitted is not appropriate for publication in the journal.

Reviewer 3 Report

Thank you for the opportunity to review this protocol paper.  It is well written with little grammar/spelling issues.

The findings of this proposed research will assist with the overall assessment of response by the locale to COVID-19, which then can be used for future public health emergencies related to the virus.  Other countries can learn from each other in this regard.

I believe the protocol be well written, with exception to the methods section of the manuscript.  Here, much more information regarding the mixed methods to be used, as delineated by the proposed secondary data set needs to be discussed.  For instance, simply stating that inferential analysis will be conducted on the data, yet not mapping data sets to study objectives/hypotheses is not sufficient.  What data sets are available to the researchers, and how are they specifically going to be analyzed?

Please map the study objectives and hypotheses to the specific data set(s) in an attempt to better address proposed qualitative and quantitative analyses to be conducted.  This is the primary part of the manuscript that will make it unique as compared to any other study protocol language in the manuscript.

Thank you.

Reviewer 4 Report

The subject of this paper is very interesting. In other words, research on COVID-19 in certain regions of South Africa is currently a very timely topic of interest.

However, this study is a kind of incomplete study, and the overall paper structure is very rudimentary. The following points should be reflected in future studies.
-Related research should be reinforced.
-1.1. Definitions of Concepts part is unnecessary.
-'3.  Results' part is the most important part of this paper, and specific results should be presented.
-'4.  Discussion' part should be described in detail after Chapter 3 is completed.

Round 2

Reviewer 4 Report

Although this study lacks analysis based on statistical data, I think it is suitable for protocol type papers.